

# Efficient phase-space generation for hadron collider event simulation

Enrico Bothmann[1], Taylor Childers[2], Walter Giele[3], Florian Herren[3],
Stefan Höche[3], Joshua Isaacson[3], Max Knobbe[1] and Rui Wang[2]

**1** Institut für Theoretische Physik, Georg-August-Universität Göttingen,
37077 Göttingen, Germany
**2** Argonne National Laboratory, Lemont, IL, 60439, USA
**3** Fermi National Accelerator Laboratory, Batavia, IL 60510, USA

## Abstract

We present a simple yet efficient algorithm for phase-space integration at hadron colliders. Individual mappings consist of a single t-channel combined with any number of s-channel decays, and are constructed using diagrammatic information. The factorial growth in the number of channels is tamed by providing an option to limit the number of s-channel topologies. We provide a publicly available, parallelized code in C++ and test its performance in typical LHC scenarios.

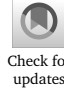
---

## Contents

---

# 1 Introduction

The problem of phase-space integration is omnipresent in particle physics. Efficient methods to evaluate phase-space integrals are needed in order to predict cross sections and decay rates for a variety of experiments, and they are required for both theoretical calculations and event simulation. In many cases, the integrand to be evaluated features a number of narrow peaks, corresponding to the resonant production of unstable massive particles. In other cases, the integrand has intricate discontinuities, arising from cuts to avoid the singular regions of scattering matrix elements in theories with massless force carriers, such as QED and QCD. In most interesting scenarios, the phase-space is high dimensional, such that analytic integration is ruled out, and Monte-Carlo (MC) integration becomes the only viable option.

Many techniques have been devised to deal with this problem [1–10]. Among the most successful ones are factorization based approaches [1–3] and multi-channel integration techniques [11]. They allow to map the structure of the integral to the diagrammatic structure of the integrand. For scalar theories, and ignoring the effect of phase-space cuts, this corresponds to an ideal variable transformation. Realistic multi-particle production processes are much more complex, both because of the non-scalar nature of most of the elementary particles, and because of phase-space restrictions. Adaptive Monte-Carlo methods [12–17] are therefore used by most theoretical calculations and event generators to map out structures of the integrand which are difficult to predict. More recently, neural networks have emerged as a promising tool for this particular task [18–25].

In this letter, we introduce a novel phase-space integrator which combines several desirable features of different existing approaches while still remaining relatively simple. In particular, we address the computational challenges discussed in a number of reports of the HEP Software Foundation [26, 27] and the recent Snowmass community study [28], which emphasize the importance of portable computing models. Our algorithm is based on the highly successful integration techniques employed in MCFM [29–32], combined with a standard recursive approach for s-channel topologies as used in many modern simulation programs. We provide a stand-alone implementation, which we call CHILI (Common High-energy Integration LIbrary),[1] which includes the Vegas algorithm [12] and MPI parallelization. We also implement Python bindings via nanobind [33] and to Tensorflow [34], providing an interface the normalizing-flow based neural network integration frameworks iFlow [20] and MADNIS [22]. To assess the performance of our new code, we combine it with the matrix element generators in the general-purpose event generator SHERPA [8, 35] and devise a proof of concept for the computation of real-emission next-to-leading order corrections by adding a forward branching generator which makes use of the phase-space mappings of the Catani-Seymour dipole subtraction formalism [36, 37].

The outline of the paper is as follows: Section 2 discusses the algorithms used in our new generator. Section 3 presents performance measures obtained in combination with COMIX [8], and Amegic [38], and Sec. 4 includes a summary and outlook.

# 2 The algorithm

One of the most versatile approaches to phase-space integration for high-energy collider experiments is to employ the factorization properties of the $n$-particle phase-space integral [3]. Consider a $2 \rightarrow n$ scattering process, where we label the incoming particles by $a$ and $b$ and outgoing particles by $1 \dots n$. The corresponding $n$-particle differential phase-space element

---

[1]The source code can be found at https://gitlab.com/spice-mc/chili.

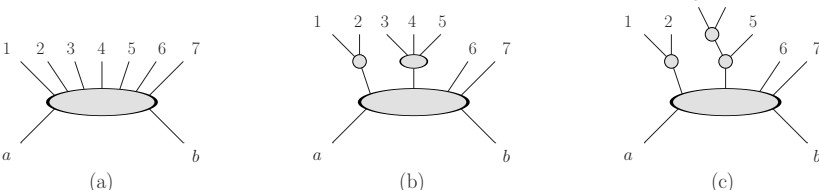

Figure 1: Example application of the phase-space factorization formula, Eq. (2). Particles 1 through 7 are produced in the collision of particles $a$ and $b$. Figure (a) represents a pure t-channel configuration, cf. Sec. 2.1. In Fig. (b), the differential 7-particle phase-space element is factorized into the production of four particles, two of which are the pseudo-particles $\{1,2\}$ and $\{3,4,5\}$, which subsequently decay. In Fig. (c), the decay of $\{3,4,5\}$ is again factorized into two consecutive decays.

reads

$$\mathrm{d}\Phi_n(a,b;1,\dots,n) = \left[\prod_{i=1}^n \frac{\mathrm{d}^3\vec{p}_i}{(2\pi)^3\,2E_i}\right](2\pi)^4\delta^{(4)}\left(p_a+p_b-\sum_{i=1}^n p_i\right). \qquad (1)$$

Following Ref. [1], the full differential phase-space element can be reduced to lower-multiplicity differential phase-space elements as follows:

$$\mathrm{d}\Phi_n(a,b;1,\dots,n) = \mathrm{d}\Phi_{n-m+1}(a,b;\pi,m+1,\dots,n)\,\frac{\mathrm{d}s_\pi}{2\pi}\,\mathrm{d}\Phi_m(\pi;1,\dots,m), \qquad (2)$$

where $\pi$ indicates an intermediate pseudo-particle of virtuality $s_\pi = p_\pi^2$. Equation (2) allows to compose the full differential phase-space element from building blocks which correspond to a single t-channel production process and a number of s-channel decays, as depicted in Fig. 1. By repeated application of Eq. (2), all decays can be reduced to two-particle decays, with differential phase-space elements $\mathrm{d}\Phi_2$. This allows to match the structure of the phase-space integral onto the structure of the Feynman diagrams in the integrand at hand, a technique that is known as diagram-based integration.

## 2.1 The t- and s-channel building blocks

In this subsection, we first describe the techniques to perform the integration using a pure t-channel differential phase-space element, $\mathrm{d}\Phi_n(a,b;1,\dots,n)$. The final-state momenta $p_1$ through $p_n$ can be associated with on-shell particles, or they can correspond to intermediate pseudo-particles whose virtuality is an additional integration variable. We start with the single-particle differential phase-space element in Eq. (1). It can be written in the form

$$\frac{\mathrm{d}^3\vec{p}_i}{(2\pi)^3\,2E_i} = \frac{1}{16\pi^2}\,\mathrm{d}p_{i,\perp}^2\,\mathrm{d}y_i\,\frac{\mathrm{d}\phi_i}{2\pi}, \qquad (3)$$

where $p_{i,\perp}$, $y_i$ and $\phi_i$ are the transverse momentum, rapidity and azimuthal angle of momentum $i$ in the laboratory frame, respectively. Many experimental analyses at hadron colliders require cuts on the transverse momentum and rapidity of jets and other analysis objects, which are easily implemented in this parametrization, leading to an excellent efficiency of the integration algorithm.

The remaining task is to implement the delta function in Eq. (1). This is achieved by combining the integral over one of the momenta, say $p_n$, with the integration over the light-cone momentum fractions used to convolute the partonic cross section with the PDFs. We

obtain

$$dx_a dx_b \, d\Phi_n(a, b; 1, \ldots, n) = \frac{dP_+ dP_-}{s} \left[ \prod_{i=1}^{n-1} \frac{1}{16\pi^2} \, dp_{i,\perp}^2 \, dy_i \, \frac{d\phi_i}{2\pi} \right]$$
$$\times \frac{d^4 p_n}{(2\pi)^3} \, \delta(p_n^2 - s_n) \Theta(E_n) \, (2\pi)^4 \delta^{(4)}\left( p_a + p_b - \sum_{i=1}^{n-1} p_i - p_n \right), \tag{4}$$

where $s$ is the hadronic center-of-mass energy, and $P_\pm = P_0 \pm P_z$ is defined using $P = \sum_{i=1}^{n-1} p_i$. Changing the integration variables from $P_+$ and $P_-$ to $s_n$ and $y_n$, it is straightforward to evaluate the delta functions, and we obtain the final expression

$$dx_a dx_b \, d\Phi_n(a, b; 1, \ldots, n) = \frac{2\pi}{s} \left[ \prod_{i=1}^{n-1} \frac{1}{16\pi^2} \, dp_{i,\perp}^2 \, dy_i \, \frac{d\phi_i}{2\pi} \right] dy_n . \tag{5}$$

This form of the differential phase-space element is particularly suited for the production of electroweak vector bosons ($W$, $Z$ and $\gamma$) in association with any number of jets. However, it may not be optimal for phase-space generation when there are strong hierarchies in transverse momenta of the jets, that may be better described by phase-space mappings similar to Fig. 1 (c).

The differential decay phase-space elements occurring in Fig. 1 (b) and (c) are easily composed from the corresponding expressions for two-body decays. In the frame of a time-like momentum $P$, this differential phase-space element can be written as

$$d\Phi_2(\{1,2\}; 1, 2) = \frac{1}{16\pi^2} \frac{\sqrt{(p_1 P)^2 - p_1^2 P^2}^3}{((p_1 P)(p_1 p_2) - p_1^2 (p_2 P)) P^2} \, d\cos\theta_1^{(P)} d\phi_1^{(P)} . \tag{6}$$

Typically, this is evaluated in the center-of-mass frame of the combined momentum, $p_1 + p_2$, where it simplifies to

$$d\Phi_2(\{1,2\}; 1, 2) = \frac{1}{16\pi^2} \frac{\sqrt{(p_1 p_2)^2 - p_1^2 p_2^2}}{(p_1 + p_2)^2} \, d\cos\theta_1^{\{1,2\}} d\phi_1^{\{1,2\}} . \tag{7}$$

Equations (5) and (7) form the basic building blocks of our algorithm.

## 2.2 The multi-channel

An optimal integrator for a particular squared Feynman diagram would be composed of a combination of the t-channel map in Eq. (5) and potentially a number of s-channel maps in Eq. (7), as sketched for various configurations in Fig. 1. The complete integrand will almost never consist of a single Feynman diagram squared, and it is therefore more appropriate to combine various such integrators in order to map out different structures in the full integrand.[2] Each of those mappings is conventionally called a phase-space "channel", and each channel is a valid phase-space integrator in it's own right. They can be combined using the multi-channel technique, which was introduced in [11]. We refer the reader to the original publication for the details of this method. Here we will briefly describe how the individual channels are constructed in our integrator.

We begin by extracting the three-particle vertices from the interaction model. Given a set of external flavors, we can use the vertex information to construct all possible topologies of

---

[2]An alternative option is to partition the integrand into terms which exhibit the structure of an individual diagram [6].

Feynman diagrams with the maximum number of propagators. For each topology, we apply the following algorithm: If an s-channel propagator is found, we use the factorization formula, Eq. (2) to split the differential phase-space element into a production and a decay part. This procedure starts with the external states and it is repeated until no more factorization is possible. As the number of possible s-channel topologies grows factorially in many cases, our algorithm provides an option to limit the maximum number of s-channels that are implemented. This helps to tailor the integrator to the problem at hand and allows to control the computational complexity. Throughout the paper, we will refer to including the maximum number of s-channels as CHILI and limiting the results to the minimum number of allowed s-channels (1 for $W$ and $Z$ processes and 0 otherwise) as CHILI (basic).

Following standard practice, we generate the virtuality of the intermediate s-channel pseudo-particles using a Breit-Wigner distribution if the particle has a mass and width, or following a $\mathrm{d}s/s^\alpha$ distribution ($\alpha < 1$), if the particle is massless. The transverse momenta in Eq. (5) are generated according to $\mathrm{d}p_\perp^2/(2p_{\perp,c} + p_\perp)^2$, where $p_{\perp,c}$ is an adjustable parameter that can be used to maximize efficiency, e.g. by setting it to the jet transverse momentum cut. The rapidities in Eq. (5) and the angles in Eq. (7) are generated using a flat prior distribution. The virtuality ($s$) for an intermediate resonance following a Breit-Wigner distribution can be generated for a particle of mass $M$ and width $\Gamma$ for an invariant mass squared between $s_{\min}$ and $s_{\max}$ with random number $r \in [0, 1)$ by

$$s = M^2 + M\Gamma \tan\left(y_{\min} + r\left(y_{\max} - y_{\min}\right)\right), \tag{8}$$

where we have defined $y_{\min,\max} = \arctan\left[\left(s_{\max,\min} - M^2\right)/(M\Gamma)\right]$.

## 2.3 Next-to-leading order calculations and dipole mappings

The integration of real-emission corrections in next-to-leading order QCD or QED calculations poses additional challenges for a phase-space integration algorithm. In order to achieve a local cancellation of singularities, subtraction methods are typically employed in these calculations [36, 39]. This makes the behavior of the integrand less predictable than at leading order, and therefore complicates the construction of integration channels. Various approaches have been devised to deal with the problem. We adopt a solution that is based on the on-shell momentum mapping technique used in the Catani-Seymour dipole subtraction scheme [36, 37] and that has long been used in generators such as MCFM [31, 32, 40] and MUNICH [41].[3]

Following Ref. [36], there are four different types of local infrared subtraction terms that are used to make real-emission corrections and virtual corrections in NLO calculations separately infrared finite. They are classified according to the type of collinear divergence (initial state or final state) and the type of color spectator parton (initial state or final state). The massless on-shell phase-space mapping for the final-final configuration (FF) reads

$$\mathrm{d}\Phi_n^{(\mathrm{FF})}(a, b; 1, \ldots, n) = \mathrm{d}\Phi_{n-1}(a, b; 1, \ldots, \widetilde{ij}, \ldots, \tilde{k}, \ldots, n)\frac{2\tilde{p}_{ij}\tilde{p}_k}{16\pi^2}\,\mathrm{d}y_{ij,k}\mathrm{d}\tilde{z}_i\,\frac{\mathrm{d}\phi}{2\pi}\left(1 - y_{ij,k}\right), \tag{9}$$

where

$$p_i^\mu = \tilde{z}_i\,\tilde{p}_{ij}^\mu + (1 - \tilde{z}_i)\,y_{ij,k}\,\tilde{p}_k^\mu + k_\perp^\mu, \qquad p_k^\mu = (1 - y_{ij,k})\,\tilde{p}_k^\mu, \qquad p_j^\mu = \tilde{p}_{ij} + \tilde{p}_k - p_i - p_k, \tag{10}$$

and where $k_\perp^2 = -\tilde{z}_i(1 - \tilde{z}_i)y_{ij,k}\,2\tilde{p}_{ij}\tilde{p}_k$ is determined by the on-shell conditions.
The massless on-shell phase-space mapping for the final-initial and initial-final configurations (FI/IF) reads

$$\mathrm{d}\Phi_n^{(\mathrm{FI/IF})}(a, b; 1, \ldots, n) = \mathrm{d}\Phi_{n-1}(\tilde{a}, b; 1, \ldots, \widetilde{ij}, \ldots, n)\frac{2\tilde{p}_{ij}p_a}{16\pi^2}\,\mathrm{d}\tilde{z}_i\mathrm{d}x_{ij,a}\,\frac{\mathrm{d}\phi}{2\pi}, \tag{11}$$

---

[3]We make this feature available only for use within SHERPA, but a future version of our stand-alone code will support it as well.

where

$$p_i^\mu = \tilde{z}_i\,\tilde{p}_{ij}^\mu + (1-\tilde{z}_i)\,\frac{1-x_{ij,a}}{x_{ij,a}}\,\tilde{p}_a^\mu + k_\perp^\mu\,, \qquad p_a^\mu = \frac{1}{x_{ij,a}}\,\tilde{p}_a^\mu\,, \qquad p_j^\mu = \tilde{p}_{ij} - \tilde{p}_a + \tilde{p}_a - \tilde{p}_i\,, \quad (12)$$

and where $k_\perp^2 = -\tilde{z}_i(1-\tilde{z}_i)(1-x_{ij,a})/x_{ij,a}\,2\tilde{p}_{ij}\tilde{p}_a$.

The massless on-shell phase-space mapping for the initial-initial configurations (II) reads

$$d\Phi_n^{(\mathrm{II})}(a,b;1,\ldots,n) = d\Phi_{n-1}(\tilde{a\imath},b;\tilde{1},\ldots,\tilde{n})\,\frac{2p_a p_b}{16\pi^2}\,d\tilde{v}_i dx_{i,ab}\,\frac{d\phi}{2\pi}\,, \qquad (13)$$

where

$$p_i^\mu = \frac{1-x_{i,ab}-\tilde{v}_i}{x_{i,ab}}\,\tilde{p}_a^\mu + \tilde{v}_i\,p_b^\mu + k_\perp^\mu\,, \qquad p_a^\mu = \frac{1}{x_{i,ab}}\,\tilde{p}_{ai}^\mu\,,$$

$$p_j^\mu = \Lambda^\mu_{\ \nu}(K,\tilde{K})\tilde{p}_j^\nu\,, \qquad \forall\, j \in \{1,\ldots,n\}\,, \quad j \neq i\,, \qquad (14)$$

and where $k_\perp^2 = -(1-x_{i,ab}-\tilde{v})/x_{ij,a}\,\tilde{v}_i\,2\tilde{p}_{ai}p_b$. The transformation, $\Lambda^\mu_{\ \nu}(K,\tilde{K})$, is defined in Sec. 5.5 of Ref. [36]. The three above mappings are sufficient to treat any real-emission correction in massless QCD. We infer the possible dipole configurations from the flavor structure of the process and combine all possible mappings into a multi-channel integrator [11].

## 2.4 Combination with normalizing-flow based integrators

With the development of modern machine learning methods, new techniques for adaptive Monte-Carlo integration have emerged, which are based on the extension [42, 43] of a non-linear independent components estimation technique [44, 45], also known as a normalizing flow. They have been used to develop integration algorithms based on existing multi-channel approaches [19, 21, 22, 25]. One of the main obstacles to scaling such approaches to high multiplicity has been the fact that the underlying phase-space mappings are used as individual mappings in a multi-channel phase-space generator. The channel selection requires additional hyperparameters, which increases the dimensionality of the optimization problem. Here we propose a different strategy. We observe that the basic t-channel integration algorithm implementing Eq. (5) requires the minimal amount of random numbers, and shows a good efficiency (cf. Sec. 3). It is therefore ideally suited to provide a basic mapping of the $n$-particle phase-space at hadron colliders into a $3n-4+2$ dimensional unit hypercube, required for combination with normalizing-flow based integrators. We provide Python bindings in CHILI via nanobind [33] and a dedicated Tensorflow [34] interface. This allows the use of the iFlow [20] and MADNIS [22] frameworks to test this idea, and to evaluate the performance of this novel algorithm.

## 3 Performance benchmarks

In this section we present first numerical results obtained with our new integrator, CHILI. We have interfaced the new framework with the general-purpose event generator SHERPA [35, 46, 47], which is used to compute the partonic matrix elements and the parton luminosity with the help of COMIX [8] and Amegic [38]. To allow performance tests from low to high particle multiplicity, we use Comix' default method of sampling of the QCD color space [8, 48], unless explicitly stated otherwise. This affects the convergence rate, and we note that better MC uncertainties could in principle be obtained for color-summed computations, but at the cost of much larger computing time at high multiplicity. The performance comparison between

Table 1: Relative Monte-Carlo uncertainties, $\Delta\sigma/\sigma$, and unweighting efficiencies, $\eta$, in leading-order calculations. The center-of-mass energy is $\sqrt{s} = 14$ TeV, jets are defined using the anti-$k_T$ algorithm with $p_{\perp,j} = 30$ GeV and $|y_j| \leq 6$. Vegas grids and multi-channel weights have been adapted using 1.2M non-zero phase-space points. For details see the main text.

| Process | SHERPA | | CHILI | | CHILI (basic) | |
|---|---|---|---|---|---|---|
| | $\Delta\sigma/\sigma$ 6M pts | $\eta$ 100 evts | $\Delta\sigma/\sigma$ 6M pts | $\eta$ 100 evts | $\Delta\sigma/\sigma$ 6M pts | $\eta$ 100 evts |
| $W^+$+1j | 0.5‰ | $7 \times 10^{-2}$ | 0.6‰ | $9 \times 10^{-2}$ | 0.6‰ | $9 \times 10^{-2}$ |
| $W^+$+2j | 1.2‰ | $9 \times 10^{-3}$ | 1.1‰ | $2 \times 10^{-2}$ | 1.2‰ | $1 \times 10^{-2}$ |
| $W^+$+3j | 2.0‰ | $1 \times 10^{-3}$ | 2.0‰ | $4 \times 10^{-3}$ | 2.9‰ | $2 \times 10^{-3}$ |
| $W^+$+4j | 3.7‰ | $2 \times 10^{-4}$ | 4.9‰ | $7 \times 10^{-4}$ | 6.0‰ | $3 \times 10^{-4}$ |
| $W^+$+5j | 7.2‰ | $4 \times 10^{-5}$ | 22‰ | $1 \times 10^{-5}$ | 26‰ | $1 \times 10^{-5}$ |
| Process | SHERPA | | CHILI | | CHILI (basic) | |
| | $\Delta\sigma/\sigma$ 6M pts | $\eta$ 100 evts | $\Delta\sigma/\sigma$ 6M pts | $\eta$ 100 evts | $\Delta\sigma/\sigma$ 6M pts | $\eta$ 100 evts |
| $Z$+1j | 0.4‰ | $2 \times 10^{-1}$ | 0.5‰ | $1 \times 10^{-1}$ | 0.5‰ | $1 \times 10^{-1}$ |
| $Z$+2j | 0.8‰ | $2 \times 10^{-2}$ | 0.8‰ | $3 \times 10^{-2}$ | 1.0‰ | $2 \times 10^{-2}$ |
| $Z$+3j | 1.3‰ | $4 \times 10^{-3}$ | 1.6‰ | $7 \times 10^{-3}$ | 2.5‰ | $4 \times 10^{-3}$ |
| $Z$+4j | 2.2‰ | $8 \times 10^{-4}$ | 3.6‰ | $1 \times 10^{-3}$ | 5.0‰ | $6 \times 10^{-4}$ |
| $Z$+5j | 3.7‰ | $1 \times 10^{-4}$ | 11‰ | $1 \times 10^{-4}$ | 13‰ | $2 \times 10^{-4}$ |
| Process | SHERPA | | CHILI | | CHILI (basic) | |
| | $\Delta\sigma/\sigma$ 6M pts | $\eta$ 100 evts | $\Delta\sigma/\sigma$ 6M pts | $\eta$ 100 evts | $\Delta\sigma/\sigma$ 6M pts | $\eta$ 100 evts |
| $h$+1j | 0.4‰ | $2 \times 10^{-1}$ | 0.4‰ | $2 \times 10^{-1}$ | 0.4‰ | $2 \times 10^{-1}$ |
| $h$+2j | 0.8‰ | $2 \times 10^{-2}$ | 0.6‰ | $5 \times 10^{-2}$ | 0.6‰ | $5 \times 10^{-2}$ |
| $h$+3j | 1.4‰ | $3 \times 10^{-3}$ | 0.9‰ | $2 \times 10^{-2}$ | 0.9‰ | $2 \times 10^{-2}$ |
| $h$+4j | 2.4‰ | $6 \times 10^{-4}$ | 1.6‰ | $6 \times 10^{-3}$ | 1.7‰ | $7 \times 10^{-3}$ |
| $h$+5j | 4.5‰ | $1 \times 10^{-4}$ | 3.2‰ | $1 \times 10^{-3}$ | 3.6‰ | $1 \times 10^{-3}$ |
| Process | SHERPA | | CHILI | | CHILI (basic) | |
| | $\Delta\sigma/\sigma$ 6M pts | $\eta$ 100 evts | $\Delta\sigma/\sigma$ 6M pts | $\eta$ 100 evts | $\Delta\sigma/\sigma$ 6M pts | $\eta$ 100 evts |
| $t\bar{t}$+0j | 0.6‰ | $1 \times 10^{-1}$ | 0.6‰ | $1 \times 10^{-1}$ | 0.6‰ | $1 \times 10^{-1}$ |
| $t\bar{t}$+1j | 0.9‰ | $2 \times 10^{-2}$ | 0.6‰ | $6 \times 10^{-2}$ | 0.9‰ | $3 \times 10^{-2}$ |
| $t\bar{t}$+2j | 1.4‰ | $4 \times 10^{-3}$ | 0.9‰ | $2 \times 10^{-2}$ | 1.4‰ | $1 \times 10^{-2}$ |
| $t\bar{t}$+3j | 2.6‰ | $7 \times 10^{-4}$ | 1.5‰ | $7 \times 10^{-3}$ | 2.9‰ | $2 \times 10^{-3}$ |
| $t\bar{t}$+4j | 4.0‰ | $1 \times 10^{-4}$ | 3.2‰ | $1 \times 10^{-3}$ | 3.5‰ | $8 \times 10^{-4}$ |
| Process | SHERPA | | CHILI | | CHILI (basic) | |
| | $\Delta\sigma/\sigma$ 6M pts | $\eta$ 100 evts | $\Delta\sigma/\sigma$ 6M pts | $\eta$ 100 evts | $\Delta\sigma/\sigma$ 6M pts | $\eta$ 100 evts |
| $\gamma$+1j | 0.4‰ | $2 \times 10^{-1}$ | 0.6‰ | $1 \times 10^{-1}$ | 0.6‰ | $1 \times 10^{-1}$ |
| $\gamma$+2j | 1.1‰ | $7 \times 10^{-3}$ | 2.2‰ | $3 \times 10^{-3}$ | 3.7‰ | $1 \times 10^{-3}$ |
| $\gamma$+3j | 2.4‰ | $5 \times 10^{-4}$ | 4.9‰ | $4 \times 10^{-4}$ | 10‰ | $1 \times 10^{-4}$ |
| $\gamma$+4j | 5.0‰ | $7 \times 10^{-5}$ | 20‰ | $3 \times 10^{-5}$ | 30‰ | $4 \times 10^{-5}$ |
| $\gamma$+5j | 9.3‰ | $2 \times 10^{-5}$ | 28‰ | $7 \times 10^{-6}$ | 36‰ | $2 \times 10^{-6}$ |
| Process | SHERPA | | CHILI | | CHILI (basic) | |
| | $\Delta\sigma/\sigma$ 6M pts | $\eta$ 100 evts | $\Delta\sigma/\sigma$ 6M pts | $\eta$ 100 evts | $\Delta\sigma/\sigma$ 6M pts | $\eta$ 100 evts |
| 2jets | 0.6‰ | $5 \times 10^{-2}$ | 0.4‰ | $1 \times 10^{-1}$ | 0.5‰ | $7 \times 10^{-2}$ |
| 3jets | 1.2‰ | $5 \times 10^{-3}$ | 1.0‰ | $1 \times 10^{-2}$ | 1.8‰ | $7 \times 10^{-3}$ |
| 4jets | 2.5‰ | $5 \times 10^{-4}$ | 2.0‰ | $3 \times 10^{-3}$ | 3.4‰ | $1 \times 10^{-3}$ |
| 5jets | 4.7‰ | $9 \times 10^{-5}$ | 5.1‰ | $6 \times 10^{-4}$ | 8.1‰ | $2 \times 10^{-4}$ |
| 6jets | 7.0‰ | $2 \times 10^{-5}$ | 15‰ | $5 \times 10^{-5}$ | 14‰ | $4 \times 10^{-5}$ |

Table 2: Relative Monte-Carlo uncertainties, $\Delta\sigma/\sigma$, and unweighting efficiencies, $\eta$, in leading-order calculations for boosted event topologies. The center-of-mass energy is $\sqrt{s} = 14$ TeV, jets are defined using the anti-$k_T$ algorithm with $p_{\perp,j} = 30$ GeV and $|y_j| \le 6$. We require a leading jet at $p_{\perp,j1} \ge 300$ GeV. Vegas grids and multi-channel weights have been adapted using 1.2M non-zero phase-space points. For details see the main text.

| Process | SHERPA | | CHILI | | CHILI (basic) | |
|---|---|---|---|---|---|---|
| | $\Delta\sigma/\sigma$ | $\eta$ | $\Delta\sigma/\sigma$ | $\eta$ | $\Delta\sigma/\sigma$ | $\eta$ |
| boosted | 6M pts | 100 evts | 6M pts | 100 evts | 6M pts | 100 evts |
| $W^++2$j | 1.4‰ | $4 \times 10^{-3}$ | 1.4‰ | $8 \times 10^{-3}$ | 2.5‰ | $2 \times 10^{-3}$ |
| $W^++3$j | 2.5‰ | $9 \times 10^{-4}$ | 3.8‰ | $6 \times 10^{-4}$ | 6.9‰ | $2 \times 10^{-4}$ |
| $W^++4$j | 4.2‰ | $2 \times 10^{-4}$ | 10‰ | $7 \times 10^{-5}$ | 17‰ | $4 \times 10^{-5}$ |
| $W^++5$j | 7.2‰ | $4 \times 10^{-5}$ | 27‰ | $3 \times 10^{-6}$ | 48‰ | $4 \times 10^{-6}$ |

| Process | SHERPA | | CHILI | | CHILI (basic) | |
|---|---|---|---|---|---|---|
| | $\Delta\sigma/\sigma$ | $\eta$ | $\Delta\sigma/\sigma$ | $\eta$ | $\Delta\sigma/\sigma$ | $\eta$ |
| boosted | 6M pts | 100 evts | 6M pts | 100 evts | 6M pts | 100 evts |
| $Z+2$j | 1.0‰ | $9 \times 10^{-3}$ | 1.1‰ | $1 \times 10^{-2}$ | 1.8‰ | $6 \times 10^{-3}$ |
| $Z+3$j | 1.6‰ | $2 \times 10^{-3}$ | 2.5‰ | $2 \times 10^{-3}$ | 5.0‰ | $5 \times 10^{-4}$ |
| $Z+4$j | 2.8‰ | $4 \times 10^{-4}$ | 7.6‰ | $2 \times 10^{-4}$ | 27‰ | $6 \times 10^{-5}$ |
| $Z+5$j | 4.6‰ | $9 \times 10^{-5}$ | 15‰ | $3 \times 10^{-5}$ | 33‰ | $2 \times 10^{-5}$ |

| Process | SHERPA | | CHILI | | CHILI (basic) | |
|---|---|---|---|---|---|---|
| | $\Delta\sigma/\sigma$ | $\eta$ | $\Delta\sigma/\sigma$ | $\eta$ | $\Delta\sigma/\sigma$ | $\eta$ |
| boosted | 6M pts | 100 evts | 6M pts | 100 evts | 6M pts | 100 evts |
| $h+2$j | 1.1‰ | $8 \times 10^{-3}$ | 0.7‰ | $4 \times 10^{-2}$ | 0.7‰ | $3 \times 10^{-2}$ |
| $h+3$j | 1.8‰ | $2 \times 10^{-3}$ | 1.0‰ | $1 \times 10^{-2}$ | 1.1‰ | $1 \times 10^{-2}$ |
| $h+4$j | 3.0‰ | $4 \times 10^{-4}$ | 1.7‰ | $3 \times 10^{-3}$ | 1.6‰ | $4 \times 10^{-3}$ |
| $h+5$j | 4.8‰ | $9 \times 10^{-5}$ | 4.2‰ | $7 \times 10^{-4}$ | 3.1‰ | $1 \times 10^{-4}$ |

| Process | SHERPA | | CHILI | | CHILI (basic) | |
|---|---|---|---|---|---|---|
| | $\Delta\sigma/\sigma$ | $\eta$ | $\Delta\sigma/\sigma$ | $\eta$ | $\Delta\sigma/\sigma$ | $\eta$ |
| boosted | 6M pts | 100 evts | 6M pts | 100 evts | 6M pts | 100 evts |
| $\gamma+2$j | 1.4‰ | $4 \times 10^{-3}$ | 2.3‰ | $2 \times 10^{-3}$ | 2.3‰ | $2 \times 10^{-3}$ |
| $\gamma+3$j | 2.3‰ | $7 \times 10^{-4}$ | 4.3‰ | $4 \times 10^{-4}$ | 9.0‰ | $1 \times 10^{-4}$ |
| $\gamma+4$j | 4.0‰ | $2 \times 10^{-4}$ | 9.9‰ | $1 \times 10^{-4}$ | 25‰ | $1 \times 10^{-5}$ |
| $\gamma+5$j | 7.3‰ | $2 \times 10^{-5}$ | 36‰ | $1 \times 10^{-6}$ | 49‰ | $3 \times 10^{-6}$ |

| Process | SHERPA | | CHILI | | CHILI (basic) | |
|---|---|---|---|---|---|---|
| | $\Delta\sigma/\sigma$ | $\eta$ | $\Delta\sigma/\sigma$ | $\eta$ | $\Delta\sigma/\sigma$ | $\eta$ |
| boosted | 6M pts | 100 evts | 6M pts | 100 evts | 6M pts | 100 evts |
| $t\bar{t}+1$j | 1.0‰ | $1 \times 10^{-2}$ | 0.7‰ | $4 \times 10^{-2}$ | 1.5‰ | $1 \times 10^{-2}$ |
| $t\bar{t}+2$j | 2.0‰ | $1 \times 10^{-3}$ | 1.1‰ | $1 \times 10^{-2}$ | 2.3‰ | $2 \times 10^{-3}$ |
| $t\bar{t}+3$j | 3.2‰ | $4 \times 10^{-4}$ | 1.9‰ | $3 \times 10^{-3}$ | 3.7‰ | $8 \times 10^{-4}$ |
| $t\bar{t}+4$j | 4.9‰ | $1 \times 10^{-4}$ | 3.8‰ | $7 \times 10^{-4}$ | 8.4‰ | $2 \times 10^{-4}$ |

| Process | SHERPA | | CHILI | | CHILI (basic) | |
|---|---|---|---|---|---|---|
| | $\Delta\sigma/\sigma$ | $\eta$ | $\Delta\sigma/\sigma$ | $\eta$ | $\Delta\sigma/\sigma$ | $\eta$ |
| $m_{jj}$ cut | 6M pts | 100 evts | 6M pts | 100 evts | 6M pts | 100 evts |
| $h+2$j | 0.9‰ | $1 \times 10^{-2}$ | 0.8‰ | $1 \times 10^{-2}$ | 0.9‰ | $1 \times 10^{-2}$ |
| $h+3$j | 1.9‰ | $1 \times 10^{-3}$ | 1.2‰ | $5 \times 10^{-3}$ | 1.3‰ | $4 \times 10^{-3}$ |
| $h+4$j | 4.1‰ | $2 \times 10^{-4}$ | 1.8‰ | $2 \times 10^{-3}$ | 2.3‰ | $1 \times 10^{-3}$ |
| $h+5$j | 16‰ | $5 \times 10^{-5}$ | 5.0‰ | $2 \times 10^{-4}$ | 4.5‰ | $5 \times 10^{-4}$ |

Table 3: Relative Monte-Carlo uncertainties, $\Delta\sigma/\sigma$, and cut efficiencies, $\varepsilon_{\text{cut}}$, in next-to-leading order calculations. The center-of-mass energy is $\sqrt{s} = 14$ TeV, jets are defined using the anti-$k_T$ algorithm with $p_{\perp,j} = 30$ GeV and $|y_j| \leq 6$. The superscript $^\dagger$ indicates a factor 10 reduction in the number of points to evaluate the Born-like components. The superscript $^*$ indicates a factor 10 reduction in the number of points to evaluate the Born-like components and the usage of a global $K$-factor as a stand-in for the finite virtual corrections.

| Process | SHERPA | | CHILI (basic) | | Process | SHERPA | | CHILI (basic) | |
|---|---|---|---|---|---|---|---|---|---|
| 1M pts | $\Delta\sigma/\sigma$ | $\varepsilon_{\text{cut}}$ | $\Delta\sigma/\sigma$ | $\varepsilon_{\text{cut}}$ | 1M pts | $\Delta\sigma/\sigma$ | $\varepsilon_{\text{cut}}$ | $\Delta\sigma/\sigma$ | $\varepsilon_{\text{cut}}$ |
| $W^+$+1j / B-like | 1.3‰ | 43% | 1.4‰ | 99% | $h$+1j / B-like | 1.3‰ | 56% | 0.7‰ | 99% |
| R-like | 4.1‰ | 46% | 3.6‰ | 58% | R-like | 3.0‰ | 52% | 2.1‰ | 69% |
| $W^+$+2j / B-like | 2.2‰ | 37% | 4.4‰ | 99% | $h$+2j / B-like | 2.6‰ | 34% | 1.4‰ | 99% |
| R-like | 1.4% | 74% | 1.5% | 80% | R-like | 8.1‰ | 68% | 8.2‰ | 87% |
| $W^+$+3j$^\dagger$/ B-like | 2.8% | 33% | 3.5% | 97% | $h$+3j$^*$/ B-like | 2.3% | 29% | 1.0% | 96% |
| R-like | 3.0% | 75% | 4.3% | 87% | R-like | 2.0% | 65% | 2.0% | 83% |

| Process | SHERPA | | CHILI (basic) | | Process | SHERPA | | CHILI (basic) | |
|---|---|---|---|---|---|---|---|---|---|
| 1M pts | $\Delta\sigma/\sigma$ | $\varepsilon_{\text{cut}}$ | $\Delta\sigma/\sigma$ | $\varepsilon_{\text{cut}}$ | 1M pts | $\Delta\sigma/\sigma$ | $\varepsilon_{\text{cut}}$ | $\Delta\sigma/\sigma$ | $\varepsilon_{\text{cut}}$ |
| $t\bar{t}$+0j / B-like | 0.4‰ | 99% | 0.8‰ | 99% | 2jets / B-like | 1.5‰ | 34% | 0.7‰ | 99% |
| R-like | 0.2‰ | 99% | 0.3‰ | 99% | R-like | 8.3‰ | 76% | 4.3‰ | 89% |
| $t\bar{t}$+1j / B-like | 1.7‰ | 61% | 1.7‰ | 99% | 3jets / B-like | 4.2% | 9.6% | 6.1‰ | 88% |
| R-like | 5.8‰ | 82% | 5.9‰ | 92% | R-like | 4.5% | 56% | 3.7% | 81% |
| $t\bar{t}$+2j / B-like | 1.5% | 45% | 1.0% | 98% | 4jets$^*$/ B-like | 4.8% | 12% | 3.2% | 90% |
| R-like | 1.4% | 78% | 1.7% | 85% | R-like | 4.7% | 50% | 3.7% | 79% |

SHERPA and CHILI would, however, be unaffected. We use the NNPDF 3.0 PDF set [49] at NNLO precision, and the corresponding settings of the strong coupling, i.e. $\alpha_s(m_z) = 0.118$ and running to 3-loop order. Light quarks, charm and bottom quarks are assumed to be massless, and we set $m_t = 173.21$. The electroweak parameters are determined in the complex mass scheme using the inputs $\alpha(m_Z) = 1/128.8$, $m_W = 80.385$, $m_Z = 91.1876$, $m_h = 125$ and $\Gamma_W = 2.085$, $\Gamma_Z = 2.4952$. We assume incoming proton beams at a hadronic center-of-mass energy of $\sqrt{s} = 14$ TeV. To implement basic phase-space cuts, we reconstruct jets using the anti-$k_T$ jet algorithm [50] with $R = 0.4$ in the implementation of FastJet [51] and require $p_{\perp,j} \geq 30$ GeV and $|y_j| \leq 6$. Photons are isolated from QCD activity based on Ref. [52] with $\delta_0 = 0.4$, $n = 2$ and $\epsilon_\gamma = 2.5\%$ and are required to have $p_{\perp,\gamma} \geq 30$ GeV. All results presented in this section are obtained with a scalable version of our new integrator using parallel execution on CPUs with the help of MPI.

Table 1 shows a comparison between MC uncertainties and event generation efficiencies in leading-order calculations, obtained with the recursive phase-space generator in COMIX and with CHILI. A brief description of the recursive phase-space integrator implemented in Comix is given in App. B. To improve the convergence of the integrals we use the Vegas [12] algorithm, which is implemented independently in both SHERPA and CHILI. The MC uncertainties are given after optimizing the adaptive integrator with 1.2 million non-zero phase-space points and evaluation of the integral with 6 million non-zero phase-space points. We employ the definition of event generation efficiency in Ref. [21], and we evaluate it using 100 replicas of datasets leading to 100 unweighted events each. For more details on our definition of event generation efficiency see App. A. We test the production of $W^+$ and $Z$ bosons with leptonic decay, on-shell Higgs boson production, top-quark pair production, direct photon production and pure QCD jet production. These processes are omnipresent in background simulations at the Large Hadron Collider (LHC), and are typically associated with additional light jet activity due to the large phase-space. Accordingly, we test the basic process with up to four additional light jets, where all additional radiated jets are assumed to be purely from QCD interactions and do

Table 4: Relative Monte-Carlo uncertainties, $\Delta\sigma/\sigma$, and unweighting efficiencies, $\eta$, in color-summed leading-order calculations. The center-of-mass energy is $\sqrt{s} = 14$ TeV, jets are defined using the anti-$k_T$ algorithm with $p_{\perp,j} = 30$ GeV and $|y_j| \le 6$. For details see the main text.

| Process (color sum) | SHERPA $\Delta\sigma/\sigma$ 6M pts | $\eta$ 100 evts | CHILI $\Delta\sigma/\sigma$ 6M pts | $\eta$ 100 evts | CHILI +NF $\Delta\sigma/\sigma$ 6M pts | $\eta$ 100 evts |
|---|---|---|---|---|---|---|
| $W^+$+1j | 0.4‰ | $2 \times 10^{-1}$ | 0.5‰ | $2 \times 10^{-1}$ | 0.2‰ | $4 \times 10^{-1}$ |
| $W^+$+2j | 0.9‰ | $2 \times 10^{-2}$ | 0.7‰ | $4 \times 10^{-2}$ | 0.7‰ | $5 \times 10^{-2}$ |
| $Z$+1j | 0.4‰ | $3 \times 10^{-1}$ | 0.4‰ | $2 \times 10^{-1}$ | 0.1‰ | $5 \times 10^{-1}$ |
| $Z$+2j | 0.7‰ | $4 \times 10^{-2}$ | 0.7‰ | $5 \times 10^{-2}$ | 0.6‰ | $6 \times 10^{-2}$ |
| $h$+1j | 0.2‰ | $4 \times 10^{-1}$ | 0.2‰ | $5 \times 10^{-1}$ | 0.05‰ | $8 \times 10^{-1}$ |
| $h$+2j | 0.6‰ | $6 \times 10^{-2}$ | 0.3‰ | $1 \times 10^{-1}$ | 0.3‰ | $2 \times 10^{-1}$ |
| $t\bar{t}$+0j | 0.2‰ | $5 \times 10^{-1}$ | 0.1‰ | $6 \times 10^{-1}$ | 0.05‰ | $7 \times 10^{-1}$ |
| $t\bar{t}$+1j | 0.5‰ | $1 \times 10^{-1}$ | 0.2‰ | $3 \times 10^{-1}$ | 0.3‰ | $2 \times 10^{-1}$ |
| $\gamma$+1j | 0.3‰ | $4 \times 10^{-1}$ | 0.7‰ | $2 \times 10^{-1}$ | 0.1‰ | $5 \times 10^{-1}$ |
| $\gamma$+2j | 1.0‰ | $1 \times 10^{-2}$ | 1.9‰ | $5 \times 10^{-3}$ | 1.4‰ | $9 \times 10^{-3}$ |
| 2jets | 0.4‰ | $2 \times 10^{-1}$ | 0.2‰ | $4 \times 10^{-1}$ | 0.08‰ | $6 \times 10^{-1}$ |
| 3jets | 0.8‰ | $2 \times 10^{-2}$ | 0.6‰ | $6 \times 10^{-2}$ | 0.7‰ | $3 \times 10^{-2}$ |

Table 5: Time for optimization and event generation in leading-order calculations. The center-of-mass energy is $\sqrt{s} = 14$ TeV, jets are defined using the anti-$k_T$ algorithm with $p_{\perp,j} = 30$ GeV and $|y_j| \leq 6$. The optimization step consists of 0.8 or 1.2 million non-zero events and the generation consists of 6 million non-zero, weighted events. All codes are generated using dual-socket eight-core Intel E5-2650v2 "Ivy Bridge" (2.6 GHz) CPUs. The CHILI +NF results are obtained using a single threaded version of SHERPA and 16 cores for the optimization of the NF parameters in Tensorflow. To be consistent with CHILI +NF, the results for SHERPA and CHILI are total runtime, summing over all MPI ranks.

| Process | SHERPA | | CHILI | | CHILI +NF | |
|---|---|---|---|---|---|---|
| (color | Opt | Gen | Opt | Gen | Opt | Gen |
| sum) | 0.8M pts | 6M pts | 0.8M pts | 6M pts | 1.2M pts | 6M pts |
| $W^+$+1j | 2m | 10m | 1m | 8m | 5m | 8m |
| $W^+$+2j | 14m | 1.9h | 13m | 1.7h | 29m | 1.3h |
| Process | SHERPA | | CHILI | | CHILI +NF | |
| (color | Opt | Gen | Opt | Gen | Opt | Gen |
| sum) | 0.8M pts | 6M pts | 0.8M pts | 6M pts | 1.2M pts | 6M pts |
| $Z$+1j | 2m | 19m | 2m | 14m | 7m | 15m |
| $Z$+2j | 30m | 3.9h | 20m | 3.3h | 58m | 2.9h |

| Process | SHERPA | | CHILI | | CHILI +NF | |
|---|---|---|---|---|---|---|
| (color | Opt | Gen | Opt | Gen | Opt | Gen |
| sum) | 0.8M pts | 6M pts | 0.8M pts | 6M pts | 1.2M pts | 6M pts |
| $h$+1j | 1m | 10m | 1m | 7m | 4m | 8m |
| $h$+2j | 8m | 1.1h | 6m | 52m | 18m | 46m |
| Process | SHERPA | | CHILI | | CHILI +NF | |
| (color | Opt | Gen | Opt | Gen | Opt | Gen |
| sum) | 0.8M pts | 6M pts | 0.8M pts | 6M pts | 1.2M pts | 6M pts |
| $t\bar{t}$+0j | 1m | 9m | 1m | 6m | 4m | 7m |
| $t\bar{t}$+1j | 6m | 54m | 6m | 42m | 17m | 40m |

| Process | SHERPA | | CHILI | | CHILI +NF | |
|---|---|---|---|---|---|---|
| (color | Opt | Gen | Opt | Gen | Opt | Gen |
| sum) | 0.8M pts | 6M pts | 0.8M pts | 6M pts | 1.2M pts | 6M pts |
| $\gamma$+1j | 2m | 15m | 1m | 11m | 6m | 13m |
| $\gamma$+2j | 22m | 2.9h | 19m | 2.2h | 38m | 2.0h |
| Process | SHERPA | | CHILI | | CHILI +NF | |
| (color | Opt | Gen | Opt | Gen | Opt | Gen |
| sum) | 0.8M pts | 6M pts | 0.8M pts | 6M pts | 1.2M pts | 6M pts |
| 2jets | 6m | 47m | 5m | 37m | 14m | 34m |
| 3jets | 27m | 3.6h | 24m | 3.0h | 45m | 2.6h |

not include additional electroweak bosons. In single boson production we do not include the trivial process without any light jets. We observe that the performance of our new integrator is well comparable to that of the recursive phase-space generator in SHERPA, especially for less than 5 additional jets with the exception of $\gamma$+jets. In many cases it shows slightly higher unweighting efficiencies. This is both encouraging and somewhat surprising, given the relative simplicity of our new approach, which does not make use of repeated t-channel factorization. Due to the uniform jet cuts, we even obtain similar performance when using the minimal number of s-channel parametrizations, where the minimal number is 1 for $W$ and $Z$ processes

and 0 otherwise. This setup is labeled as CHILI (basic) in Tab. 1. The results suggest that a single phase-space parametrization may in many cases be sufficient to compute cross sections and generate events at high precision, which is advantageous in terms of computing time and helps to scale the computation to higher multiplicity processes. Moreover, it circumvents the problems related to multi-channel integration discussed in [21,22] when combining our integrator with neural network based adaptive random number mapping techniques. We note that this configuration is also used by MCFM [29].

Table 2 shows a similar comparison as in Tab. 1, but in addition we apply a cut on the leading jet, requiring $p_{\perp,j1} > 300$ GeV. This configuration tests the regime where the hard system receives a large boost, and there is usually a strong hierarchy between the jet transverse momenta. In these scenarios we expect the complete CHILI integrator to outperform the basic configuration with a t-channel only, which is confirmed by the comparison in Tab. 2. The only exception to this is the $\gamma + 5j$ process, which may be a result of the poor integration accuracy from CHILI for this process. The lower right sub-table shows a configuration where we do not apply the additional transverse momentum cut, but instead use a large di-jet invariant mass cut, typical for VBF searches and measurements, $m_{j1,j2} \geq 600$ GeV. Here we see that CHILI and CHILI (basic) are roughly comparable and perform better than the default SHERPA integrator, with the exception of $h + 2j$.

Table 3 shows a comparison of MC uncertainties and cut efficiencies for various next-to-leading order QCD computations. We use the Catani-Seymour dipole subtraction method [36], where the value of an arbitrary infrared-safe observable, $O$, can be computed with the help of the Born differential cross section, $B$, the UV renormalized virtual corrections, $V$, the collinear mass factorization counterterms, $C$, and a set of differential and integrated infrared subtraction counterterms, $D_i$ and $I_i = \int d\Phi_{+1} D_i$, where $d\Phi_{+1}$ is the differential one-emission phase-space associated with the production of an additional parton [36]:

$$
\begin{aligned}
\langle O \rangle = &\int d\Phi_n \left[ B(\Phi_n) + V(\Phi_n) + C(\Phi_n) + \sum_i I_i(\Phi_n) \right] O(\Phi_n) \\
&+ \int d\Phi_{n+1} \left[ R(\Phi_{n+1}) O(\Phi_{n+1}) - \sum_i D_i(\Phi_{n+1}) O(\Phi_{n,i}) \right].
\end{aligned}
\tag{15}
$$

We note that in the second integral, the value of the observable, $O$, is computed based on the real-emission phase-space point in the first term, and based on the projected Born-like phase-space points in the dipole subtration terms. Each dipole term has its own, specific projection. The fact that the cancelation of infrared enhancements in the second integral occurs non-locally in phase-space makes the evaluation particularly cumbersome with Monte-Carlo methods. While the associated integral is finite for any infrared safe observable, it typically has large Monte-Carlo uncertainties due to imperfect cancelations of positive and negative contributions.

We assign the shorthand B-like for first line in Eq. (15), and the shorthand R-like for the second line. Both calculations exhibit different structures than at leading order in QCD, cf. [40]. The real-emission integrals in particular test the efficiency of the dipole mapping described in Sec. 2.3, which is designed to match the structure of the differential infrared counterterms, $D_i$. It can be seen that our new algorithm has a much better cut efficiency than the recursive phase-space generator in SHERPA, which is again advantageous in terms of overall computing time. The cut efficiency, $\varepsilon_{cut}$ is defined as the ratio between the number of Monte-Carlo points that pass the phase-space cuts, and the total number of points. The MC uncertainty for a given number of phase-space points is reduced at low jet multiplicity, and generally comparable to the recursive phase-space generator. Given the simplicity of the CHILI approach, this is a very encouraging result for the development of NLO simulations on modern computing architectures. If a speedup of the matrix element calculation is obtained, for example through analytic

expressions [53], accelerated numerical evaluation [54–57] or the usage of surrogate methods [58, 59], then the linear scaling with the number of outgoing particles of the basic CHILI generator at leading order, and the polynomial scaling with the number of outgoing particles of the dipole-based generator,[4] will become an important feature.

Table 4 shows a comparison of the Vegas-based CHILI integrator and the neural-network assisted integrator for color summed matrix elements. We use the single channel configuration of MADNIS [22] (which is consistent with iFlow [20]) in combination with the basic CHILI integrator, while the Vegas-based version of CHILI includes all possible $s$-channel mappings. The network is setup with 6 rational quadratic spline coupling layers [43] with random permutations, each consisting of a neural network with 2 layers with 16 nodes each using a leaky ReLU activation function. In general, a coupling layer invertibly maps an input vector onto another one. For an $n$-dimensional input vector $x$, let $A$ and $B$ denote two disjoint sets of $\{1, \ldots, d\}$. Then the coupling layer mapping is defined via

$$
\begin{aligned}
x^A &\mapsto y^A := x^A, \\
x^B &\mapsto y^B := C\left(x^B; m(x^A)\right),
\end{aligned}
\tag{16}
$$

where $m$ is any function and $C$ is a separable, invertible function on $\mathbb{R}^{|B|} \times m(x^A)$. Here $|B|$ denotes the cardinality of $B$, i.e. the number of dimensions in the set $B$, and separability means that the mapping is applied element-wise as

$$
C(x^B; m(x^A)) = \left(C_1\left(x_1^B; m(x^A)\right), \ldots, C_{|B|}\left(x_{|B|}^B; m(x^A)\right)\right)^T.
\tag{17}
$$

The splitting of the input vector into two disjoint sets, where one set is being mapped and the other set is used to parameterize the mapping combined with the requirements on the function $C$, allows for a simple computation of the Jacobian determinant and crucially does not require the inversion of $m$. This allows the usage of neural networks as functions $m$. For more details see e.g. Refs. [19, 20]

The network is trained using 20 epochs of training with 100 batches of 1000 events per epoch with the variance as the loss term as in Ref. [22]. The learning rate starts at 0.001 and decays each epoch by $l_0/(1 + l_d s/d_s)$, where $l_0$ is the initial learning rate, $l_d = 0.01$ is the decay rate, $s$ is the number of steps, and $d_s = 100$ is the number of steps before applying the decay. Optimizing these parameters to achieve peak performance is beyond the scope of this project and can be done in a similar fashion as in Ref. [21].

The timings for SHERPA, CHILI, and CHILI +NF are given in Tab. 5. Here we find that the optimization of the normalizing flows is significantly slower than the Vegas optimization, even when including all of the possible phase-space channels. However, the generation of 6 million weighted events is approximately the same for the lowest multiplicity processes, but the normalizing flow approach does better after adding one additional jet. This is a combination of having a better cut efficiency and significantly fewer channels. We leave a more detailed investigation on the timing benefits to unweighting efficiency benefits at high multiplicities to a future work.

Figures 2 and 3 show the weight distributions from 6 million phase-space points after training for the simplest and next to simplest of our test processes. We compare the recursive integrator of COMIX, CHILI with Vegas and CHILI in combination with MADNIS. All results have been computed using color summed matrix elements. It can be seen that the normalizing flow based integrator yields a very narrow weight distribution in most cases. A narrow weight distribution leads to good unweighting efficiencies as shown in Tab. 4. However, the

---

[4]The number of dipole subtraction terms in the Catani-Seymour method scales at most as $n^3$ with the number $n$ of external partons.

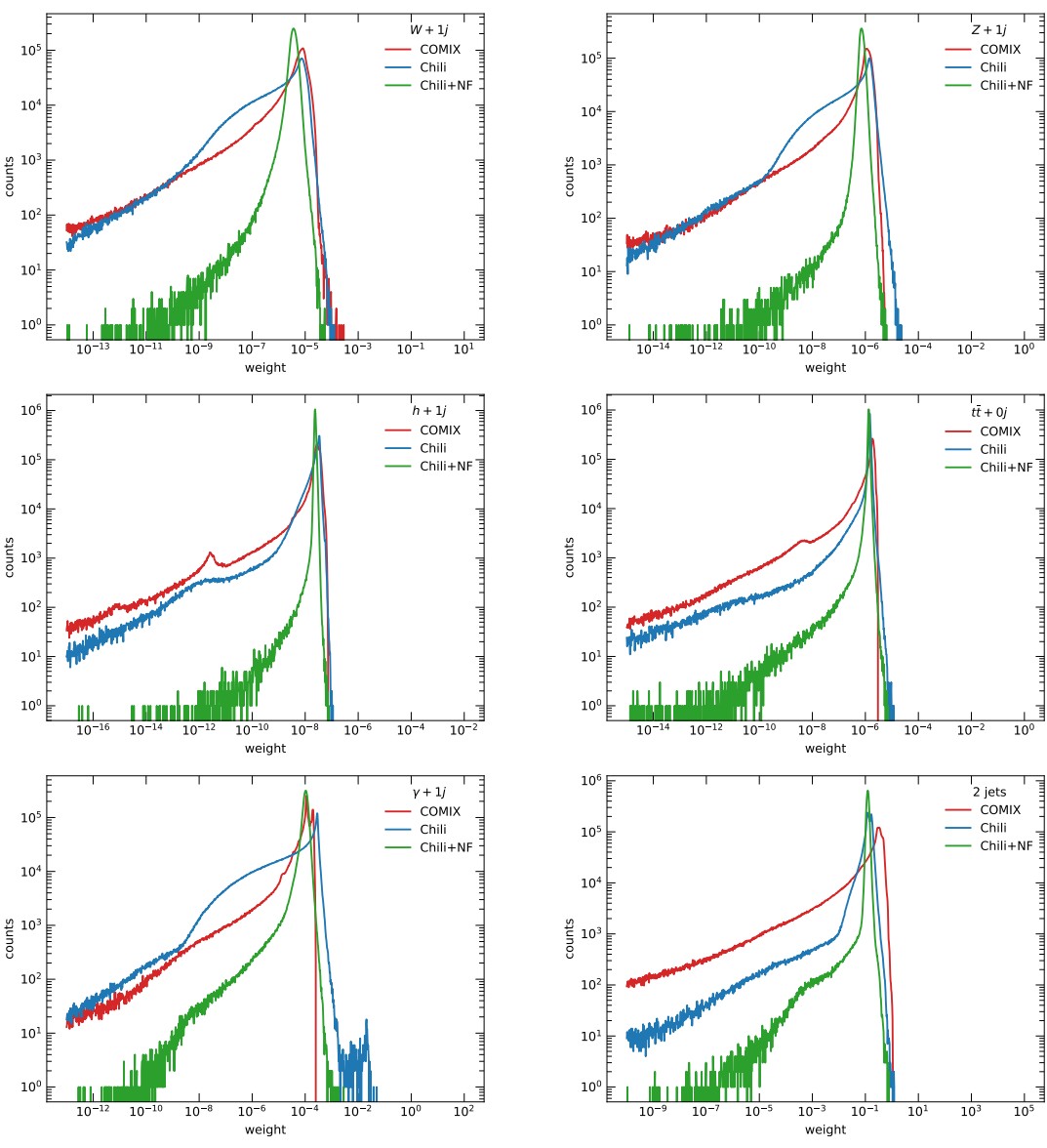

Figure 2: Weight distribution for the lowest multiplicity processes found in Tab. 4. Each curve contains 6 million events. The COMIX integrator is shown in red, the CHILI with Vegas is shown in blue, and CHILI with normalizing flows is shown in green. The results for $W+1j$ is in the upper right, $Z+1j$ in the upper left, the middle row consists of $h+1j$ and $t\bar{t}+0j$, and the bottom row has $\gamma+1j$ and dijets respectively.

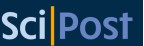

Figure 3: Same as Fig. 2, but with an additional jet for each process.

default COMIX integrator leads to a sharper upper edge of the weight distribution in the more complex scenarios of Fig. 3, which is more favorable for unweighting. This indicates that the multi-channel approach with additional s-channels is favorable at high multiplicities. We will investigate further this effect using the technology developed in Ref. [22]. Furthermore, while the variance loss is optimal for achieving a narrow weight distribution, by attempting to minimize the variance of the weight distribution. However, this tends to result in a symmetric distribution about the mean. This in turn leads to a less sharp upper edge in the weight distribution, which results in a sub-optimal unweighting efficiency. Additionally, the number of points required to reach optimal performance for the normalizing flow is significantly higher than the Vegas based approaches, as demonstrated in Ref. [20]. A study of the effect on the choice of loss function and other hyper-parameters involved in the normalizing flow approach is left to a future work to improve the unweighting efficiency at higher multiplicities and the convergence of the integrator.

## 4 Outlook

We have presented a new phase-space generator that combines various existing techniques for hadron collider phase-space integration into a simple and efficient algorithm. This new integrator is not meant to become a replacement of the existing, tried and tested techniques in state of the art parton-level event generators. Instead, we aimed at a simple, yet practical solution with good Monte-Carlo efficiency, that offers the possibility to build a scalable framework for event generation and can easily be ported to computing architectures other than CPUs. Our new algorithm satisfies this requirement, because its computational complexity scales linearly or at most polynomially with the number of external particles, and the complexity of the mapping can easily be adapted to the problem at hand. We have implemented the method in a scalable framework for CPU computing. Several extensions of this framework are in order: It should be ported to allow the usage of GPUs. Computing platforms other than CPUs and GPUs could be enabled with the help of Kokkos [60] or similar computing models. This becomes particularly relevant in light of recent advances in computing matrix elements on GPUs using portable programming models [54–57]. In addition, the techniques for real-emission corrections should be extended beyond SHERPA, in order to make our generator applicable to a wider range of problems. We also plan to further explore the combination of our new techniques with existing neural-network based integration methods.

## Acknowledgments

We thank John Campbell for many stimulating discussions and his support of the project.

**Funding information**   This research was supported by the Fermi National Accelerator Laboratory (Fermilab), a U.S. Department of Energy, Office of Science, HEP User Facility. Fermilab is managed by Fermi Research Alliance, LLC (FRA), acting under Contract No. DE–AC02–07CH11359. The work of F.H., S.H. and J.I. was supported by the U.S. Department of Energy, Office of Science, Office of Advanced Scientific Computing Research, Scientific Discovery through Advanced Computing (SciDAC) program, grant "HPC framework for event generation at colliders". F.H. acknowledges support by the Alexander von Humboldt foundation. E.B. and M.K. acknowledge support from BMBF (contract 05H21MGCAB). Their research is funded by the Deutsche Forschungsgemeinschaft (DFG, German Research Foundation) – 456104544; 510810461.

## A   Phase-space efficiency

Classical Monte Carlo unweighting relies on finding the maximum weight $w_{\max}$ during an inital optimization phase. Thereafter, every Monte Carlo weight is compared against this maximum weight in a procedure called unweighting. However, the procedure is prone to outliers in the weight distribution with the potential to drastically reduce the unweighting efficiency and thus also the compute efficiency. The procedure we used, as introduced in Ref. [21], aims to reduce the impact of outliers. In the following we briefly recall the algorithm:

1. For $N_{\mathrm{Opt}}$ point in the last optimization step, generate $n$ sets of events, each with $N_{\mathrm{Opt}}$ points.

2. From these events, choose $m$ times $N_{\mathrm{Opt}}$ event samples and determine the maximum weight for each of them.

3. Define $w_{\max}$ as the median of the maximum weight of each of the sets.

The numbers $n, m$ are to be chosen such that the unweighting efficiency stabilizes. In our case, we choose $n = m = 100$.

## B Recursive phase-space generator

An efficient way for phase-space generation inspired by the diagram-based techniques in [3] is given by the recursive phase-space generator introduced in [8]. It relies on a matching of the basic building blocks for the differential phase-space to the Berends-Giele recursion.

Consider the $2 \to n$ differential phase-space in Eq. (1). According to Eq. (2), it can be factorized, where $\pi = \{1, \ldots, m\}$ corresponds to a set of particle indices. If we denote a subset of all possible particle indices by greek letters, we can apply Eq. (2) repeatedly to decompose the complete phase-space into basic building blocks corresponding to the $s$-channel production factor $(2\pi)^4 \delta^4(p_\alpha + p_b - \sum_i p_i)$ and the two-body decays $d\Phi_2(\alpha, b; \pi, \{a, b, 1, \ldots, n\} \setminus \{\alpha, b, \pi\})$ and $d\Phi_2(\pi; \rho, \pi \setminus \rho)$. These objects can be matched to the three-particle vertices occurring in the tree-level matrix element, as long as the particle index $b$ is held fixed. Similarly, the integral $ds_\pi / 2\pi$, introduced in Eq. (2), can be matched to an s-channel propagator. It is then possible to show that the phase-space weight for a multi-channel integrator replicating the structures present in the Berends-Giele recursion can be computed using the same recursive algorithm.

A key advantage of this recursive phase-space generator is that the computational complexity scales at most exponentially with the number of outgoing particles, while for diagram-based algorithms it scales factorially. More details on the algorithm, including a simple example, can be found in [8].

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
