# Peer review of "Efficient phase-space generation for hadron collider event simulation"

_SciPost Physics, doi:SciPost Phys. 15, 169 (2023)_

## Round 1 · Referee Report · Anonymous · 2023-5-4

Report
In the first part, the paper describes a specific parameterization of phase space which applies to typical high-energy scattering processes at the LHC. The parameterization is combined with an integration algorithm which involves the well-known VEGAS integrator. The algorithm is implemented as library which works within the framework of the Monte-Carlo package Sherpa. The tables collect numerical results on the relative error and unweighting efficiency for several classes of LHC processes, thus comparing the new parameterization with the default Sherpa setup.
In a second part which is somewhat independent, the VEGAS integrator is replaced by an alternate integrator derived from a normalizing-flow implementation. Further numerical results quantify the effect of exchanging algorithms.
Detailed accounts of algorithms with a systematic evaluation of efficiency are rarely seen as independent papers in the Monte Carlo community. Therefore, I see good reasons for publishing this particular work. The selection of process classes includes a major fraction of typical backgrounds at the LHC, so the quoted numerical results can be considered as relevant for future reference.
However, the presentation in the text does not fit the form of a proper physics paper. In the present form, it appears as if it was intended as a conference report where many details and definitions are omitted due to lack of space, referring to the literature instead. For a proper physics paper, I would expect the text to be as self-contained as possible, within reason. The paper combines results and methods from various research areas (e.g., MC development, NLO methods, machine learning), and it cannot be assumed that the intended audience has sufficient knowledge in all of these fields. I think that the usefulness for the reader would improve if central terms and formulas are explicitly given instead of just providing references to the literature, even if this introduces redundancies for some expert readers.
Requested changes
1. Sec. IIB: Breit-Wigner resonances vs Table I: it is not clear whether the considered processes actually contain resonances where Breit-Wigner mappings apply. Are the radiated jets of pure QCD type, or do they include further electroweak resonances as intermediate states? In any case, it might be appropriate to quote the relevant mappings for pseudo-particles explicitly since they are probably one-line formulas - even if those are 'standard practice'.
2. Related to 1., in the text (not in the Table caption), it is mentioned that the W and Z particles are actually resonances with a lepton final state while the Higgs and top are on-shell. For a technical comparison, such an inconsistent treatment does not appear to be justified - if this is the only place where Breit-Wigner mappings appear, their effect is essentially irrelevant to the numerical study and should be eliminated.
3. Sec.III, 1st paragraph: 'color sampling' is a technical term which requires a proper definition or citation.
4. Sec.III, 2nd paragraph: a concise explanation of the Comix algorithm may be required at this point. If this is impractical, a precise reference to the description in the original literature would help the reader. Understanding both algorithms and their actual differences is central to understanding the paper.
5. same paragraph: The precise definition of the unweighting efficiency is essential for interpreting the numerical results. I would recommend to repeat the definition in this paper so the reader does not have to consult Ref.21.
6. same paragraph: The term 'in many cases slightly better' should be qualified. In Table I, for the largest multiplicities the original Sherpa errors appear to be significantly better instead.
7. same paragraph: the definition of CHILI(basic) is unclear. What determines a 'minimal number of s-channel parameterizations', and where does this differ from the default setup?
8. similarly, the claim that in Tab.II the complete integrator outperforms the basic one is not evident. Compare, e.g., eta for h+5j with gamma+5j. Maybe the highest-multiplicity results are unreliable? By how much do the efficiencies fluctuate?
9. next paragraph: given that various details of the Catani-Seymour subtraction method are laid out in the introduction, a vague definition of B-like and R-like in plain text only is insufficient.
10. same paragraph: 'linear scaling' and 'polynomial scaling' is unclear. I would assume that this refers to scaling in the number of external particles. Can the reason for this scaling behavior be given?
11. Table III: there is no definition for the cut efficiency (epsilon_cut) in the context of NLO/CS subtraction.
11. next paragraph: I think it is reasonable to expect a definition of 'coupling layers' (as concatenated mappings) and 'rational quadratic spline' (as invertible binned interpolation) in terms of straightforward mathematical expressions, assuming that some readers may only have a pure physics background. Furthermore, the relevance of the involved neural network should be clarified - is this network a means for finding the optimal spline parameters which define the actual mapping, or do the spline mappings actually 'consist of a neural network' as the text states? This statement appears confusing.
12. Table IV: The eta entry (col. CHILI/basic) for gamma + 2j is a typo? If not, this is an exceptional improvement in this particular case, can it be explained?
13. next paragraph: the statement 'while the variance loss ... are not significantly penalized' should be explained in some more detail.
14. The outlook section is very generic and does not capture the main results of the paper. I would recommend to rewrite this section. The conclusions should summarize and qualify the actual impact of the proposed phase-space reparameterization and of applying normalizing flows instead of VEGAS. A superficial look at the tables does not reveal improvements by orders of magnitude, so maybe the ultimate message is inconclusive? A meaningful statement on the computational cost of training vs production (comparing VEGAS and normalizing flow) would also be of general interest.
15. Refs 60 and 61: do arXiv entries exist?

---

## Round 1 · Referee Report · Anonymous · 2023-5-4

Strengths
1. This work strenghtens the link between particle physics phenomenology and modern Neural Network techniques
2. There is a clear potential for follow-up work and application of the integration algorithm presented in the paper on a larger scale
Weaknesses
1. In order to reach optimal efficiency, the hyper-parameters of the NN-assisted integrator still require tuning. Further investigation is required. Nevertheless the first results look encouraging.
Report
In this work the authors present a new algorithm for the integration of scattering cross sections and event generation at colliders. The algorithm is implemented into a publicly available C++ library named CHILI.
The topics addressed by the paper are important and timely. Indeed, improving the efficiency of phase-space integration and sampling is one of the necessary steps forward to improve predictions of multi-particle signatures which are becoming increasingly accessible at the LHC. The most popular tools used for adaptive importance sampling, such as VEGAS or other codes based on similar philosophy, are not optimal in presence of resonant peaks not aligned with the coordinate axes. This well known issue motivates the exploration of alternative strategies inspired by modern Neural Networks (NN), like generative models based on coupling layers (also known as Normalizing Flows, or NF) which are also considered in this study. However, the optimal combination of NN-based methods with multi-channels is still an open question due to the large number of hyper-parameters that one should tune.
The interesting aspect of the new integrator is that it combines several desired features from different existing approaches: (i) a multi-channel apparatus consisting of a main $t$-channel parametrization complemented by a number of $s$-channels generated using flavor information (with the option to limit the number of the latters for very complex processes); (ii) VEGAS- and NF-based adaptive importance sampling (the latter via interfaces to dedicated libraries); (iii) MPI parallelization. Feature (ii) is particularly useful as it facilitates systematic comparisons within the same framework.
The performance of the new integrator is assessed via comparisons with the default multi-channel integrators implemented in SHERPA and COMIX. A number of benchmarks processes, ordered by increasing light-jet multiplicity, is analysed for a given number of phase space points. The authors show that the new algorithm, combining the $t$-channel and a variable number of $s$-channels (labeled as "CHILI" in Tables I-IV) performs reasonably well in the considered test cases. For two classes of processes (i.e. on-shell $t\bar{t}$+jets and $h$+jets) it outperforms the results of SHERPA, while in the other cases it gives comparable results typically up to two additional light jets. Interestingly, even when used in the single $t$-channel mode (labeled "CHILI (basic)" in the paper) the new integrator shows a good performance for processes with uniform jet cuts. On the other hand, when selecting kinematical regions which enhance hierarchies among jets, the additional $s$-channels are more important as expected. Yet, the simplified "CHILI (basic)" approach keeps very good score for $t\bar{t}/h$+jets also in these kinematical regions. This is appealing as it reduces the computational complexity when interfacing to NN-assisted integration. A possible caveat is that the neglected $s$-channels could play a more prominent role when top quarks and Higgs bosons are treated as unstable particles and allowed to decay (either on-shell or off-shell). In such case the simplified "CHILI (basic)" mode might not be as optimal. While this is beyond the scope of the present letter, it should be considered for future investigations.
In the next step the authors compare the VEGAS-based "CHILI" with the NF-based "CHILI (basic)" integrator. In all the considered cases the latter outperforms the former, which demonstrates the excellent adaptability of the NF. This result looks promising for future developments and encourages applications on a larger scale. To fully assess the performance, a comparison with SHERPA for the same setup as well as a measure of the computational cost required by the three case studies would be desirable.
In the last part of the analysis, the authors focus on the differential weight distribution as obtained with COMIX, CHILI+VEGAS and CHILI+NF integrators. The CHILI+NF results exhibits the narrowest peaks for processes with one additional jet, which indicates that the integrand function is flatter. On the other hand, for larger jet multiplicities the NN-trained integrator is not yet optimal and, as also pointed out by the authors, further investigation of the hyper-parameters and optimizations are required.
Overall, I find that the paper is well written and the findings that it reports are interesting. The paper meets the criteria of SciPost Physics. I recommend this paper for publication after the points reported below are addressed.
1. At page 3 of the manuscript the following sentence is reported: "As the number of possible $s$-channel topologies grows factorially in many cases, our algorithm provides an option to limit the maximum number of $s$-channels that are implemented."
The difference between results labeled as "CHILI" and "CHILI (basic)" is that the latter employs a single $t$-channel mapping whereas the former uses additional $s$-channels. What is the number of $s$-channels effectively considered in "CHILI" results? Is it the maximal number for the process considered, or it is limited?
I ask the authors to clarify this point.
2. Eq.(4) has a typo: $\delta(p_n - s_n)$ should read $\delta(p_n^2 - s_n)$.
3. In Table IV the VEGAS-based "CHILI" and NF-based "CHILI (basic)" approaches are compared. It would be helpful to provide for comparison the SHERPA result at least for one of the processes reported (say $W^{+} + 1/2$ jets).
4. Another interesting detail is how faster/slower is the training of the NF in comparison to VEGAS adaptation (with the current hyper-parameter settings, even if not fully optimal). The statement at page 7, "Additionally, the number of points required to reach optimal performance for the normalizing flow is significantly higher than the Vegas based approaches", suggests that the NF requires more time to be trained, however I suppose that this is eventually (partially) compensated by the increased convergence rate and cut efficiency. I encourage the authors to provide in the text some indication about the CPU time required for the three cases.
5. As the authors point out in the text, the efficiency of the single $t$-channel parametrization ("CHILI (basic)" mode)
is not optimal when there are strong hierarchies in transverse momenta of the jets. However, the superior adaptability of the NF could partially compensate the absence of additional $s$-channels. It would be interesting to check (at least for one process) how the results of Table IV change when using $p_{\perp,j1} > 300$ GeV.
I leave it up to the authors to decide whether they want to make such a comparison or leave it for future work.
6. At page 7 the following sentence is reported: "However, the default Comix integrator leads to a sharper upper edge of the weight distribution in the more complex scenarios of Fig. 3, which is favorable for unweighting. This indicates that the multi-channel approach with additional $s$-channels is favorable at high multiplicities."
Given the sentence above, it is unclear whether the result labeled as "Chili" in Figures 2,3 adopts the single $t$-channel as "Chili+NF" does, or it incorporates $s$-channels (note that in Tables I-IV the label "Chili" denotes using $t$-channel and additional $s$-channels). I ask the authors to clarify this point in the text to avoid confusion.
Requested changes
At the end of my Report I have raised a list of 6 minor points that should be addressed.

---

## Round 1 · Referee Report · Anonymous · 2023-5-8

(Invited Report)- Cite as: Anonymous, Report on arXiv:2302.10449v1, delivered 2023-05-08, doi: 10.21468/SciPost.Report.7163
Strengths
1 - the authors present a simplified phase space parameterization that is tailored to hadron collider physics.
2 - the authors present useful performance numbers for typical applications, including the moderate gains from adding machine learning algorithms.
Weaknesses
1 - when comparing to COMIX, the authors choose to use the same sample sizes and do not comment on the relative CPU time consumption. It would be interesting so see if the simplicity of their algorithm gives a substantial benefit.
2 - the authors glance over the significant deterioration of their algorithms performance for higher multiplicities that can be seen in tables I and II. It is not surprising that a simpler algorithm performs worse in more complicated situations. Nevertheless, this should be weighed against the performance benefits.
Report
I keep it short, because I missed the deadline: it is a useful paper and should be published after small corrections.
Requested changes
1 - In the mass shell delta function for particle n in equation (4), p_n should read p_n^2.
2 - in the 5th line of section D, the phrase "phase space mappings are indeed multi channels" is ungrammatical and should be rephrased along the lines "... use a multi channel phase space sampling ..."
3 - in the 8th to last line on page 7, the claim of "excellent unweighting efficiencies shown in table IV" can be misunderstood as to refering to the normalizing flow mentioned in the first half of the sentence. However, table IV shows that in most cases, the gains for \eta due to NF are moderate at best. The sentence should therefore be split in two to avoid that misunderstanding.

---

## Round 2 · Referee Report · Anonymous · 2023-8-1

Strengths
1- The paper gives an excellent estimate of the real life potential of modern machine learning methods for phase space sampling.
2- The fact that the simple untuned algorithm appears to be on par with established workhorses demonstrates that the ML methods have potential, but also that the dramatic improvements expected by some in the field will still require significant additional work.
Report
The revision have improved the paper. The scope, strengths and weaknesses of the algorithm are now described more explicitely. The added tables and table columns are very helpful. This should make the paper also accessible as a reference for practitioners who are not yet experts.
I recommend the paper for publication in its present form.

---

## Round 2 · Referee Report · Anonymous · 2023-8-3

Strengths
1- This work strenghtens the link between particle physics phenomenology and modern Neural Network techniques
2- There is a potential for follow-up work and application of the new integration algorithm on a larger scale.
Report
The authors have addressed all comments in a satisfactory way. In particular the extension of Table IV and the addition of Table V are very helpful for the reader. Overall the presentation and discussion of the results is improved. I recommend publishing the manuscript in its present form.

---

## Round 2 · Author Response

Dear editors,
We would like to first of all thank the referees for their detailed responses to our work. Please find below the response to their reports. Additionally, we include a pdfdiff of the original version with the new updated version for the ease of the referees.
Report 1: 1. We have added a sentence when discussing the limits of the number of s-channels describing what is Chili and what is Chili (basic). 2. Thank you for pointing out the typo. It has now been corrected. 3. We have added the information for Sherpa for the color summed results along with the Chili and Chili+NF results. 4. We have added a new table containing the timing information (broken down into optimization time and generation time) for the same set of processes that are included in the Chili+NF comparison. 5. We have decided to leave this to a future work. 6. We have clarified this in the main text.
Report 2: 1. We mention in Section III that the W^+ and Z boson production processes include the leptonic decays. Therefore, the Breit-Wigner resonance mappings are used there. We have added the mappings to the end of the section. The additional jets are only QCD jets and no additional electroweak resonances are considered. The text has been modified to make that more clear. 2. We believe it is not inconsistent to treat the Higgs and tops as on-shell while appropriately handling the off-shell decays of the W and Z bosons. This is due to the fact that the width of the Higgs and the top are significantly smaller than the W and Z bosons, and can be well approximated by separating the calculation into a production and decay component using the narrow width approximation for the Higgs and top. 3. We added some more details on our QCD color treatment and furthermore added the relevant citations. Added text is: “To allow performance tests from low to high particle multiplicity, we use Comix’ default color-dressed sampling of the QCD color space [49], unless explicitly stated otherwise. For the same reason, color sampling is also used when using Amegic [39].” 4. We have added a brief description of the recursive phase-space generator in Comix to App.C and referred to the original publication for details. 5. We added a separate appendix for the definition of our unweighting efficiency. 6. We clarified this to talk about the slightly higher unweighting efficiency for processes with less than 5 additional jets and excluding photon+jet production. 7. We have added explicitly what the minimum number of s-channel parameterizations are for each process. 8. We point out that the complete integrator outperforms the basic one for all pieces except for the photon+5j calculation for the boosted case. This difference may be attributed to the poor integration uncertainty obtained by the Chili methods. Additionally, we made a comment about how the performance for the Higgs production with VBF cuts are similar for the complete integrator and the basic integrator. We have modified the text accordingly as well. 9. We included a brief description of the main components of NLO calculations in the dipole subtraction formalism in Sec.III 10. We have reformulated this sentence and added a footnote on the polynomial scaling with the number of external particles of the dipole-based integrator. 11. We have defined the cut efficiency in the main text. 12. We have added some definitions of coupling layers. And have clarified the goals of using the networks. 13. We thank the referee for pointing out the typo. It has been corrected. 14. This has been clarified to point to the fact that the variance loss tends to result in a symmetrical distribution, and therefore there is a less sharp upper edge in the weight distribution. The sharpness of this upper edge is directly related to the unweighting efficiency. 15. We have reformulated the conclusions and added a sentence to the introduction to clarify that the main aim of the paper is to provide a new phase-space integration algorithm that not only has good performance but also offers a solution applicable to many-core CPU and GPU computing. As such, the fact that our new integrator is comparable in performance to Sherpa, and better only in some cases, is acceptable and not particularly noteworthy. Indeed, its main feature is its simplicity, making it a portable solution. 16. We have added the preprint number for Ref. 60. There is no arXiv entry for Ref. 61
Report 3: 1. Thank you for pointing out this typo. This has been corrected 2. We agree and have rephrased the sentence to clarify how the mappings build up a multi-channel. The sentence now reads, “One of the main obstacles to scaling such approaches to high multiplicity has been the fact that the underlying phase-space mappings are used as individual mappings in a multi-channel phase-space generator.” 3. We rephrased this to make it clear that narrow weight distributions lead to good unweighting efficiencies, but highlight that a sharper upper edge is still more important.
Again, we would like to thank the referees for their insightful comments and through review of our work.
Regards, Joshua Isaacson (on behalf of the authors)

---

## Round 2 · List of Changes

The points are given above in the comments

---

## Editorial Decision

published